

# Coral reproduction in Western Australia

James Gilmour[1,2], Conrad W. Speed[1,2] and Russ Babcock[2,3]

[1] Australian Institute of Marine Science, The UWA Oceans Institute, Crawley, Western Australia, Australia
[2] Western Australian Marine Science Institution, Perth, Western Australia, Australia
[3] Commonwealth Scientific and Industrial Research Organisation, Oceans and Atmosphere, Brisbane, Queensland, Australia

Corresponding author
James Gilmour,
j.gilmour@aims.gov.au

## ABSTRACT

Larval production and recruitment underpin the maintenance of coral populations, but these early life history stages are vulnerable to extreme variation in physical conditions. Environmental managers aim to minimise human impacts during significant periods of larval production and recruitment on reefs, but doing so requires knowledge of the modes and timing of coral reproduction. Most corals are hermaphroditic or gonochoric, with a brooding or broadcast spawning mode of reproduction. Brooding corals are a significant component of some reefs and produce larvae over consecutive months. Broadcast spawning corals are more common and display considerable variation in their patterns of spawning among reefs. Highly synchronous spawning can occur on reefs around Australia, particularly on the Great Barrier Reef. On Australia's remote north-west coast there have been fewer studies of coral reproduction. The recent industrial expansion into these regions has facilitated research, but the associated data are often contained within confidential reports. Here we combine information in this grey-literature with that available publicly to update our knowledge of coral reproduction in WA, for tens of thousands of corals and hundreds of species from over a dozen reefs spanning 20° of latitude. We identified broad patterns in coral reproduction, but more detailed insights were hindered by biased sampling; most studies focused on species of *Acropora* sampled over a few months at several reefs. Within the existing data, there was a latitudinal gradient in spawning activity among seasons, with mass spawning during autumn occurring on all reefs (but the temperate south-west). Participation in a smaller, multi-specific spawning during spring decreased from approximately one quarter of corals on the Kimberley Oceanic reefs to little participation at Ningaloo. Within these seasons, spawning was concentrated in March and/or April, and October and/or November, depending on the timing of the full moon. The timing of the full moon determined whether spawning was split over two months, which was common on tropical reefs. There were few data available for non-*Acropora* corals, which may have different patterns of reproduction. For example, the massive *Porites* seemed to spawn through spring to autumn on Kimberley Oceanic reefs and during summer in the Pilbara region, where other common corals (e.g. *Turbinaria* & *Pavona*) also displayed different patterns of reproduction to the *Acropora*. The brooding corals (*Isopora* & *Seriatopora*) on Kimberley Oceanic reefs appeared to planulate during many months, possibly with peaks from spring to autumn; a similar pattern is likely on other WA reefs. Gaps in knowledge were also due to the difficulty in identifying species and issues with

methodology. We briefly discuss some of these issues and suggest an approach to quantifying variation in reproductive output throughout a year.

# INTRODUCTION

## Reproduction in scleractinian corals

Sexual recruitment underpins the maintenance of most coral communities, so knowing their peak times of reproductive output is critical to the management of human activities that reduce recruitment to the adult population. Larval production, recruitment, and early post-recruitment survival in corals are reduced by extreme variation in physical factors such as temperature and salinity (*Bassim, Sammarco & Snell, 2000*; *Harrison & Wallace, 1990*; *Harrison, 2011*; *Negri, Marshall & Heyward, 2007*) or degraded water quality (*Gilmour, 1999*; *Harrison & Ward, 2001*; *Humphrey et al., 2008*; *Markey et al., 2007*; *Negri & Heyward, 2001*). Model projections highlight the implications of prolonged reductions in larval recruitment for the maintenance of coral populations, and particularly their recovery following disturbances (*Babcock, 1991*; *Done, 1987*; *Edmunds, 2005*; *Fong & Glynn, 2000*; *Gilmour et al., 2006*; *Smith et al., 2005*). The times of reproduction also influence the community recovery via connectivity to other coral reefs (*Gilmour, Smith & Brinkman, 2009*; *Done, Gilmour & Fisher, 2015*). For example, the larvae of brooding corals are released several times a year under a range of hydrodynamic conditions, but typically disperse over relatively short distances (< several kilometres), whereas the larvae of spawning corals are produced during one or a few discrete periods, and disperse over larger distances (> several kilometres). A detailed understanding of community reproduction is therefore required to mitigate human activities around critical periods of larval production and to inform the design of management networks reliant on estimates of larval exchange (*Carson et al., 2010*; *Kool, Moilanen & Treml, 2013*).

Most scleractinian corals have one of four patterns of sexual reproduction, depending on their sexuality (hermaphroditic or gonochoric) and developmental mode (brooding or broadcast spawning) (*Baird, Guest & Willis, 2009*; *Fadlallah, 1983*; *Harrison & Wallace,1990*; *Harrison, 2011*; *Richmond & Hunter, 1990*). In brooding corals, the fertilisation of eggs and subsequent development of larvae occur within the parental polyps. Larvae are competent to settle shortly after their release from the polyp, with planulation typically occurring over several months each year. In contrast, colonies of broadcast spawning corals typically release their gametes into the water column once a year, where fertilization and larval development occur, after which larvae disperse for days to weeks before settling. Some coral species (or cryptic sub-species) have more complex patterns of reproduction (e.g. *Pocillopora damicornis*), while blurred species boundaries and flexible breeding systems continue to confound our understanding of reproduction in many coral taxa (*van Oppen et al., 2002*; *Veron, 2011*; *Willis, 1990*; *Willis et al., 2006*).

Reproductive activity in spawning corals can be remarkably synchronised, culminating in the release of gametes by a high proportion of species and colonies during a few nights each year (mass spawning), or spawning by a similar proportion of colonies and species may be protracted over many nights and several months (*Baird, Guest & Willis, 2009*; *Harrison & Wallace, 1990*; *Harrison, 2011*). The ultimate factor driving high synchrony, particularly within species, is probably successful fertilisation and larval recruitment. However, a wide range of environmental factors underlie this success and cue spawning over increasingly fine temporal scales, such as water temperature, day length, moon phases and tidal amplitude (*Baird, Guest & Willis, 2009*; *Guest et al., 2005a*; *Harrison & Wallace, 1990*; *Penland et al., 2004*; *van Woesik, 2010*). These cues all interact to synchronise spawning within communities, so it is tempting to view mass spawning as a phenomenon that occurs at the community level, whereas each species is in fact responding independently to its environment. As conditions vary, gametogenic cycles in each species will respond differently, as their environmental optima may differ or because the environment provides fewer synchronising cues (*Oliver et al., 1988*). Indeed, environmental stress will reduce the energy available for gametogenesis and the likelihood of corals reproducing during a given year (*Michalek-Wagner & Willis, 2001*; *Ward, Harrison & Hoegh-Guldberg, 2000*), also confounding generalities about spawning patterns. The species composition of reefs changes as environmental conditions vary, further influencing the patterns of reproduction at the reef scale. Clearly there is significant scope for reproduction of coral assemblages on reefs to vary regionally and depart from the 'mass spawning' discovered on the Great Barrier Reef (*Babcock et al., 1986*; *Harrison et al., 1984*) and subsequently pursued by some investigations of coral reproduction around the world. This variation in timing and synchrony results in a range of reproductive patterns, from temporal isolation of spawning species to a highly synchronous mass-spawning.

Mass spawning in scleractinian corals was first discovered on parts of the GBR in austral spring (*Harrison et al., 1984*; *Willis et al., 1985*), where it is perhaps more synchronous than on any other coral reef worldwide. However, even on the GBR there is a spatial and temporal variation in mass-spawning. For example, the near-shore reefs spawn one month earlier than those on mid- and outer-shelf reefs (*Willis et al., 1985*), while the high- and low-latitude reefs have a more protracted period of spawning at times other than during spring (*Baird, Guest & Willis, 2009*; *Baird, Marshall & Wolstenholme, 2002*; *Harrison, 2008*; *Oliver et al., 1988*; *Wilson & Harrison, 2003*). Additionally, spawning times within coral assemblages also vary among years according to the timing of the full moon within the spawning window. The date of the full moon occurs several days earlier each month than in the previous year, causing spawning times to shift periodically (e.g. from October–November) if gametes are not yet mature at the time of full moon. Similarly, when the full moon falls near the edge of the spawning window then only some colonies will have mature gametes, so spawning occurs following two consecutive full moons (e.g. October and November). This phenomenon has been termed 'split spawning' and typically occurs every few years, but can occasionally occur over consecutive years (*Baird, Guest & Willis, 2009*; *Willis, 1985*).
Many of the early studies leading to the discovery of mass spawning on the GBR involved rigorous sampling of colonies using a range of methods throughout the year, which established synchronous reproductive cycles within and among populations (*Wallace, 1985*). This led to more intensive sampling over weeks and days, which established the remarkable synchrony among many colonies and species over a few nights each year. In contrast, some subsequent studies have focused on identifying the species participating in mass spawning events but not quantifying the proportion of participating colonies or the frequency of spawning during other times (nights, weeks, months, and seasons) of the year. Without estimates of the reproductive state of colonies during other times of the year, a relative assessment of the participation in mass spawning events is not possible; if there is a low participation in the mass spawning then there is no knowledge of the other time(s) of spawning, whereas if there is a moderate to high participation then it may be assumed incorrectly that spawning during the other time(s) is negligible. For example, a rigorous sampling of the reproductive state of coral populations throughout the year has identified a second spawning by populations and even some colonies on the GBR (*Stobart, Babcock & Willis, 1992*; *Wolstenholme, 2004*) and other reefs around the world (*Dai, Fan & Yu, 2000*; *Guest et al., 2005b*; *Mangubhai, 2009*; *Mangubhai & Harrison, 2006*; *Oliver et al., 1988*). Focussing only on the participation of corals in the mass spawning can also miss the times of reproduction for entire species that are common and functionally important, such as the massive *Porites* (*Harriott, 1983a*; *Kojis & Quinn, 1982*). Additionally, brooding corals are a significant component of many reefs, and planulation in populations and colonies is typically spread over several months throughout the year (*Ayre & Resing, 1986*; *Harriott, 1992*; *Harrison & Wallace, 1990*; *Harrison, 2011*; *Tanner, 1996*).

Despite considerable research effort on the GBR, there is still not a detailed understanding of spatial and temporal variation in coral reproduction at the scale of entire assemblages. This highlights the difficulty in obtaining a similar understanding for the remote coral reefs on Australia's west coast, where far less research has been conducted. Most studies of coral reproduction in Western Australia (WA) have been conducted over a few months at several reefs, of which there are few published accounts (but see Table S1), leaving large gaps in knowledge. The gaps are significant because the existing data illustrate the unique patterns of reproduction displayed by WA coral communities and the extent to which they vary among habitats and regions. The rapid industrial expansion through regions of WA in the last decade has seen an increase in the number of studies of coral reproduction, but much of the associated data are contained within confidential reports to industry and government. Here we combine some of the information in this grey-literature with that in public reports and papers, to update our current knowledge of coral reproduction in WA. This includes data for tens of thousands of corals and hundreds of species, from over a dozen reefs spanning 20° of latitude. From these data we identify broad latitudinal patterns, but many gaps in knowledge remain due to paucity of data, biased sampling, and in some instances poor application of methodology. We therefore conclude with a brief discussion around issues of sampling design and methodology, and suggest one approach to quantifying the

significance of periods of reproductive output by coral communities, which is among the suite of information required by managers to moderate the effects of human activities along Australia's west coast.

## METHODS

### Western Australian regions and sources of reproductive data

Western Australia's coral reefs span more than 12,000 km of coastline and 20° of latitude, ranging from tropical to temperate climates, from coastal reefs to oceanic atolls hundreds of kilometres from the mainland (*Veron & Marsh, 1988*; *Wilson, 2013*). Consequently, WA has a phenomenal diversity of habitats and coral communities, with a corresponding range in reef-level patterns of coral reproduction. Because of these broad patterns in coral community composition, the examination of patterns of reproduction presented here is divided among six regions: (1) Kimberley Oceanic, (2) Kimberley, (3) Pilbara, (4) Ningaloo, (5) Abrolhos and Shark Bay, and (6) Rottnest and southwest WA (Fig. 1). Among these regions, the diversity of coral species and genera decreases with increasing latitude (Fig. 1), although coral cover can be similar among the tropical reefs and those at the subtropical Abrolhos Islands, before then decreasing in the temperate southwest (*Abdo et al., 2012*; *Johannes et al., 1983*; *McKinney, 2009*; *Richards & Rosser, 2012*; *Richards, Sampey & Marsh, 2014*; *Speed et al., 2013*; *Veron & Marsh, 1988*).

Regional data or data summaries of coral reproduction were taken from journal articles and public reports, unpublished data, and confidential reports to industry and government (Table S1). Where possible, raw data were interrogated and summaries produced across reefs for each region. However, in other instances raw data were not available and regional summaries were based on tables and text within reports that had not been peer-reviewed. Given the scope of these data, discrepancies also existed among studies and there are likely errors in data collection, analyses and species identification. Some regional summaries were adjusted to account for obvious errors in data or conclusions in some reports and the most likely patterns of reproduction were sometimes extrapolated from limited data. Additionally, samples were typically biased by factors such as the environmental conditions, the community composition, the sampling design and the methods used. For example, inferences about the patterns of reproduction on a reef were heavily biased when: data exist for a few species of *Acropora* but the community was dominated by non-*Acropora* corals that reproduce at different times; environmental stress inhibited gametogenesis causing a large portion of the assemblage not to reproduce in a period; spawning was split over two consecutive months but only one month was sampled; coral species and/or genera were incorrectly identified. The issues were most acute in studies with limited spatial and temporal replication. For these reasons, a summary of information that commonly biases inferences about patterns of coral reproduction is presented for each region, to place in context the reproductive data, and times of spawning for species were assigned a level of confidence according to the available data (Tables 1, 2 and S2).

Coral reef habitats of WA are characterised by widely contrasting environments, but all are exposed to considerable wave energy generated by seasonal cyclones and/or storms.

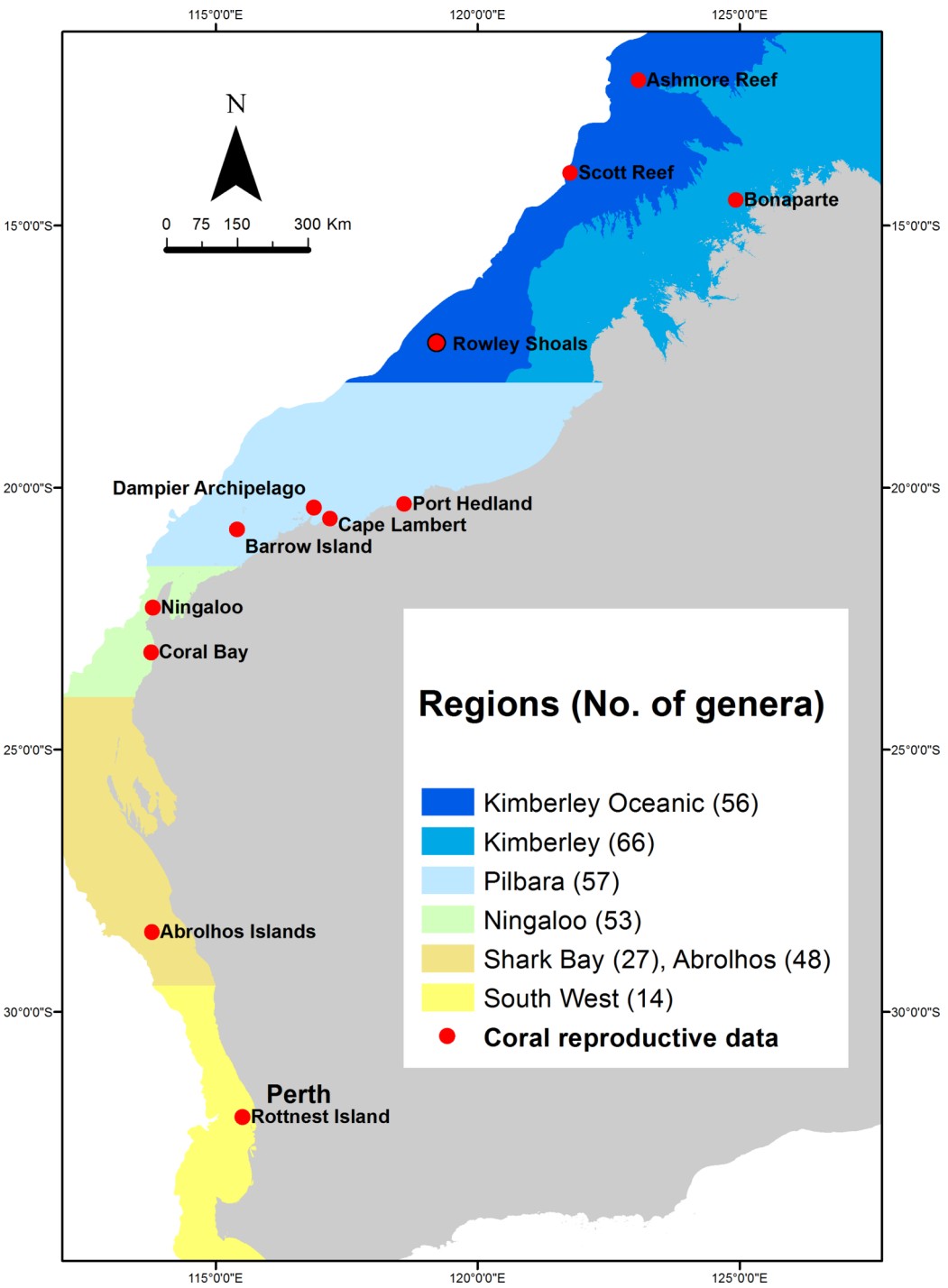

**Figure 1 Regions in which the composition of coral reefs and the proposed patterns of coral reproduction differ most significantly across Western Australia.** Numbers in brackets indicate the number of coral genera identified in each region (see Table 1). Red circles indicate reefs at which data on coral reproduction were available, from which inferences about the differences among regions were drawn.
**Table 1 Regional variation in coral diversity and reproduction across Western Australia.** The number of species within each coral genus know to occur within each region of WA, and the number for which reproductive data are available. The percentage of species within each genus known to reproduce in spring or autumn within each region, of the total sampled. Regions are colour coded according to Fig. 1.

| Region | Genus | Total known species | Number of species sampled | | Spawning % (number) of species sampled | |
|---|---|---|---|---|---|---|
| | | | Spring | Autumn | Spring | Autumn |
| Kimberley Oceanic | Acropora | 63 | 39 | 49 | 90 (35) | 94 (46) |
| | Echinophyllia | 3 | 1 | 2 | 0 (0) | 100 (2) |
| | Favia | 13 | 4 | 6 | 75 (3) | 100 (6) |
| | Favites | 8 | 3 | 3 | 33 (1) | 100 (3) |
| | Goniastrea | 6 | 2 | 6 | 100 (2) | 100 (6) |
| | Hydnophora | 4 | 2 | 1 | 50 (1) | 100 (1) |
| | Lobophyllia | 3 | 1 | 0 | 100 (1) | – |
| | Merulina | 2 | 2 | 2 | 0 (0) | 100 (2) |
| | Montipora | 28 | 0 | 0 | – | – |
| | Platygyra | 6 | 0 | 3 | – | 100 (3) |
| Kimberley | Acropora | 39 | 35 | 16 | 42 (15) | 87 (14) |
| | Echinophyllia | 3 | 0 | 0 | – | – |
| | Favia | 9 | 2 | 1 | 0 (0) | 100 (1) |
| | Favites | 6 | 2 | 1 | 0 (0) | 100 (1) |
| | Goniastrea | 7 | 4 | 1 | 0 (0) | 100 (1) |
| | Hydnophora | 4 | 4 | 0 | 75 (3) | – |
| | Lobophyllia | 2 | 2 | 0 | 0 (0) | – |
| | Merulina | 1 | 1 | 0 | 0 (0) | – |
| | Montipora | 23 | 0 | 0 | – | – |
| | Platygyra | 5 | 3 | 1 | 0 (0) | 100 (1) |
| Pilbara | Acropora | 49 | 35 | 43 | 34 (12) | 98 (42) |
| | Echinophyllia | 2 | 0 | 0 | – | – |
| | Favia | 10 | 1 | 8 | 0 (0) | 87 (7) |
| | Favites | 7 | 2 | 4 | 50 (1) | 100 (4) |
| | Goniastrea | 7 | 5 | 7 | 0 (0) | 100 (7) |
| | Hydnophora | 4 | 1 | 1 | 0 (0) | 100 (1) |
| | Lobophyllia | 3 | 1 | 2 | 0 (0) | 100 (2) |
| | Merulina | 2 | 1 | 1 | 0 (0) | 100 (1) |
| | Montipora | 28 | 4 | 3 | 0 (0) | 66 (2) |
| | Platygyra | 6 | 4 | 6 | 0 (0) | 100 (6) |
| Ningaloo | Acropora | 39 | 17 | 26 | 12 (2) | 92 (24) |
| | Echinophyllia | 2 | 2 | 2 | 0 (0) | 100 (2) |
| | Favia | 8 | 0 | 2 | – | 100 (2) |
| | Favites | 8 | 0 | 1 | – | 100 (1) |
| | Goniastrea | 7 | 1 | 1 | 0 (0) | 100 (1) |
| | Hydnophora | 4 | 1 | 1 | 0 (0) | 100 (1) |
| | Lobophyllia | 4 | 1 | 1 | 0 (0) | 100 (1) |
| | Merulina | 2 | 2 | 2 | 0 (0) | 100 (2) |
| | Montipora | 28 | 2 | 2 | 0 (0) | 100 (2) |
| | Platygyra | 6 | 1 | 2 | 0 (0) | 100 (2) |

(Continued)

| Region | Genus | Total known species | Number of species sampled | | Spawning % (number) of species sampled | |
|---|---|---|---|---|---|---|
| | | | Spring | Autumn | Spring | Autumn |
| Abrolhos | *Acropora* | 39 | 0 | 20 | – | 100 (20) |
| | *Echinophyllia* | 2 | 0 | 1 | – | 100 (1) |
| | *Favia* | 8 | 0 | 5 | – | 100 (5) |
| | *Favites* | 8 | 0 | 5 | – | 100 (5) |
| | *Goniastrea* | 7 | 0 | 2 | – | 0 (0) |
| | *Hydnophora* | 2 | 0 | 0 | – | – |
| | *Lobophyllia* | 3 | 0 | 1 | – | 100 (1) |
| | *Merulina* | 1 | 0 | 1 | – | 100 (1) |
| | *Montipora* | 26 | 0 | 4 | – | 100 (4) |
| | *Platygyra* | 2 | 0 | 1 | – | 100 (1) |
| South West | *Acropora* | 1 | 0 | 1 | – | 100(1) |
| | *Echinophyllia* | 0 | 0 | 0 | – | – |
| | *Favia* | 1 | 0 | 0 | – | – |
| | *Favites* | 4 | 0 | 0 | – | – |
| | *Goniastrea* | 3 | 0 | 2 | – | 50 (1) |
| | *Hydnophora* | 0 | 0 | 0 | – | – |
| | *Lobophyllia* | 0 | 0 | 0 | – | – |
| | *Merulina* | 0 | 0 | 0 | – | – |
| | *Montipora* | 1 | 0 | 1 | – | 100 (1) |
| | *Platygyra* | 0 | 0 | 0 | – | – |

**Note:**

Dashes lines indicate no data for that genus. Diversity data are summarised from several key references (*Berry, 1993*; *Berry & Marsh, 1986*; *Done et al., 1994*; *Richards et al., 2015*; *Richards & Rosser, 2012*; *Richards et al., 2009*; *Richards, Sampey & Marsh, 2014*; *Veron, 1993*; *Veron & Marsh, 1988*).

Reef habitats range from open ocean atolls surrounded by deep oligotrophic waters in the Kimberley Oceanic Region, to reefs heavily influenced by coastal processes such as tidally driven sediment resuspension in the inshore Kimberley and Pilbara Regions. From the coastal fringing reefs of Ningaloo, to the subtropical and temperate reefs at the Abrolhos Island and the Southwest Region, tidal processes are less extreme, waters are clearer and often lower in nutrients. This is due in part to the southward flowing Leeuwin Current which intensifies in winter, moderating winter temperature minima and assisting the transport of coral larvae to southern reefs (*Cresswell, 1996*; *D'Adamo et al., 2009*; *Hatcher, 1991*). Consequently, there is a high level of reef development in the sub-tropical reefs at the Abrolhos Islands. While the low latitude reefs in the Kimberley have the highest species diversity, they also experience the most pronounced differences in environmental conditions and community composition between the oceanic reefs and those adjacent to the mainland (*Richards et al., 2015*; *Richards, Sampey & Marsh, 2014*). Similarly, within the Pilbara Region, community composition differs between the most frequently studied inshore reefs in the Dampier Archipelago where most reproductive data exist, and mid-shelf around Barrow and Montebello islands (*Richards & Rosser, 2012*; *Richards, Sampey & Marsh, 2014*). More information about the environmental

**Table 2 Regional variation in spawning for coral species sampled most rigorously on Western Australian reefs.** Regions are colour coded according to Fig. 1.

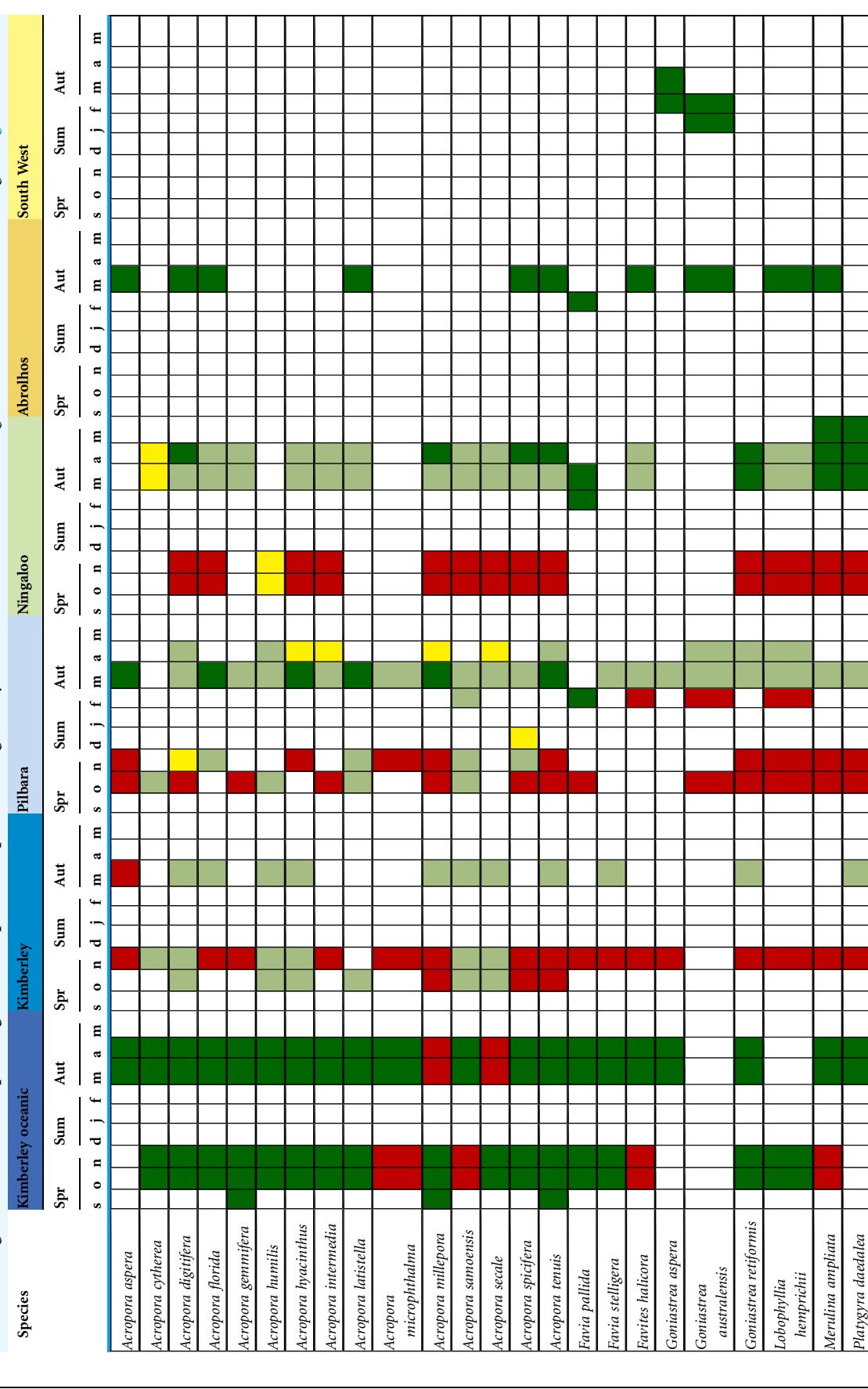

**Notes:**

Seasons and months are: Spring, Spr; September, s; October, o; November, n; Summer, Sum; December, d; January, j; February, f; Autumn, Aut; March, m; April, a; May, m. Spawning has not been recorded during Winter months (June, July, August) in Western Australia and they have been excluded. Taxonomic revisions are summarised in Table S2. Based on the available data, the sampling design and the methods used, confidence in the inferred months of spawning were ranked qualitatively according to:

■ Confident. Evidence based on the presence of pigmented eggs in colonies prior to the predicted dates of spawning in many colonies, sites and years; the presence and absence of pigmented eggs in many colonies around the predicted dates of spawning; and/or direct observations of spawning in multiple colonies.

■ Likely. Evidence based on the presence of pigmented eggs in many colonies prior to the predicted dates but with limited spatial and temporal replication; and/or most evidence indicates spawning during this month but with some contradictory data among studies.

■ Possible. Evidence based on the presence of large but unpigmented eggs several weeks prior to the predicted dates of spawning; and/or contradictory data among studies due to sampling design, methodology or species identification.

■ Unlikely. No evidence of spawning; pigmented or large unpigmented eggs absent from samples of many colonies, sites and years within several weeks of the predicted dates of spawning.

characteristics and the context for reef development and coral reproduction in each region is provided in Supplemental Information 1. Preceding the reproductive summary for reefs within each region is information to place these data in context, which includes: the species diversity and community composition of corals; the number and types of reefs, sites and species for which reproductive data were collected and the time(s) of sampling; whether colonies were affected by disturbances at the times of sampling; and the methods used to infer the times of spawning or planulae release.

## RESULTS

### Regional patterns of coral reproduction: Kimberley Oceanic

The oceanic reefs of the Kimberley are atolls rising from depths of several hundred meters, with over 300 species and 57 genera of hard corals. Coral cover in many habitats can be over 70%, and much of the remaining substrata are covered in coralline and turf algae, with a very low cover of macroalgae and other benthic organisms. The Acroporidae are typically the dominant family of hard corals, followed by the Poritidae, Faviidae and Pocilloporidae, while soft corals are also common.

Coral reproduction has been investigated at all of the Kimberley oceanic reefs during one or more years (Table S1). From Ashmore, Cartier, Scott and Seringapatam Reefs, and the Rowley Shoals, several thousand colonies from over 130 species and 30 genera have been sampled during the autumn and/or spring spawning seasons, in one or more years. Of the total number of *Acropora* species know in the region, approximately 62% were sampled in spring and 78% in autumn, compared to 20 and 32% of non-*Acropora* species, respectively (Table 1). The majority of the sampling has been conducted at Scott Reef, where there was sampling of colonies prior to the spawning in autumn and spring in consecutive years from 2007–2010, including repeated sampling of some tagged colonies. There has been comparatively little sampling at other times of year, so inferences about spawning during summer months may be underestimated. In most instances, the times of spawning were inferred from in situ ranking of gamete development, in addition to microscopic investigation of egg sizes and histological analyses of some spawning corals and brooding corals. Spawning has also been observed in situ on several occasions.

The existing data suggest that most species of corals on the oceanic atolls are broadcast spawners. Spawning has been inferred to occur primarily during spring and autumn, with a larger proportion of species and colonies participating in the autumn mass spawning than in the multi-specific spawning during spring (Tables 2 and S2). Many species participated in both spawning events, but most colonies spawn only once a year (i.e. within-population biannual spawning). Of the species of *Acropora* sampled in spring (*n* = 39) and autumn (*n* = 49), 90% were reproductively active in spring and 94% in autumn, compared to 10% in spring and 32% in autumn for the common non-*Acropora* species (*n* = 73) (Tables 2 and S2). For the species sampled repeatedly over several years, approximately 40% spawned only in autumn, less than 10% only in spring, and approximately 55% in both autumn and spring; within species, a similar proportion (> 30%) of colonies spawning during each season. A similar pattern was evident in the

additional 30 species of *Acropora* and 20 species of non-*Acropora* sampled less rigorously ($n = 5$–10 colonies yr$^{-1}$), but for a higher proportion of non-*Acropora* species and colonies spawning in autumn; *Favia stelligera* and *F. pallida* spawned during both seasons and *Diploastrea heliopora* spawned only during spring (Table S2). More intensive sampling of the non-*Acropora* species may increase the proportion of instances of within-species biannual spawning among these species.

Within each season, spawning most commonly occurred during March and October, but varied according to the timing of the full moon. Split-spawning occurred every few years during both seasons and occasionally over consecutive years; splits usually occurring between March and April in autumn, and October and November in spring, following full moons that fell in the last week or so of the preceding months. Spawning has been observed directly in autumn and/or spring during six years, and colonies were sampled before and after to check for the disappearance of pigmented eggs. Based on these observations, spawning usually occurred 7–9 nights after the full moon during neap tides. However, the times of spawning varied among years and occurred any time from the night of the full moon to around 10 days after.

The majority of corals showed evidence of spawning either in March and/or April, and October and/or November, with the exception of the massive *Porites*. At the times of sampling during autumn and spring, pigmented eggs were observed in only a few massive *Porites* colonies, but massive *Porites* can spawn eggs with comparatively little pigmentation (*Stoddart, Stoddart & Blakeway, 2012*). Histological analyses of samples collected at these times indicated that colonies were dioecious and released eggs and sperm over several months in the year from spring to autumn. A peak in reproductive activity was not obvious, and stages of gamete development indicated spawning over several months from October–May, in contrast to the peak in spawning observed in massive *Porites* on other reefs around Australia (*Kojis & Quinn, 1982*; *Stoddart, Stoddart & Blakeway, 2012*). The sampling of all species was restricted a few months each year around two main spawning events, and the extent of spawning following other lunar phases and months has not been investigated in detail. The potential exists for at least some colonies and/or species to spawn during other times. For example, a small proportion of *Acropora millepora*, *A. tenuis*, *A. polystoma*, *A. gemmifera* and *Goniastrea edwardsii* colonies at Ashmore Reef had pigmented eggs in early February or September 2011, indicating they would either spawn a month earlier than most other corals or would retain their eggs until the next month; alternately, early spawning in some corals during 2011 could reflect higher than normal water temperatures. In addition the variation in times of broadcast spawning, larval production in the brooding corals also occurs outside of the dominant spawning events. Histological analyses confirmed that *Isopora brueggemanni*, *I. palifera*, *Seriatopora hystrix* and *Stylophora pistillata* were brooding corals in the offshore Kimberley region. *Isopora brueggemanni* and *S. hystrix* were most intensively sampled and contained gametes in all stages of development and planula larvae during most months from October–May. There was no clear peak in reproductive activity in the brooding corals and larvae were apparently released over many months from spring to autumn.

## Regional patterns of coral reproduction: Kimberley

There are diverse and extensive reef systems throughout Kimberley region, including inner shelf, fringing and patch reefs, exposed platforms and subtidal banks around the coastline and islands (*Richards, Sampey & Marsh, 2014*; *Speed et al., 2013*; *Wilson, 2013*). There are over 300 species of hard corals from 71 genera, and clear cross-shelf differences in species distributions exist between the coastal and offshore locations, with 27 species (8%) recorded only from nearshore locations and 111 species (33%) recorded only at offshore locations (*Richards, Sampey & Marsh, 2014*). There are no quantitative data describing the relative abundances of corals throughout the inshore Kimberley, but qualitative descriptions highlight the considerable variation in habitats and coral assemblages. For example, leeward intertidal reefs may be characterised by branching and tabular *Acropora*; subtidal zones can have a high cover and diversity of corals dominated by massive *Porites* and species of Faviidae and foliose corals; exposed fringing reefs may have a comparatively low cover and diversity of corals dominated by massive Faviidae and soft corals; extensive tidal pools throughout the region can have a high cover and diversity of corals different to those in other zones (*INPEX, 2011*; *Wilson, 2013*).

There are very few reproductive data for coral assemblages in the inshore Kimberley region, particularly given the extent and diversity of the reefs (Table S1). Inferences of coral reproduction in the region are largely based on surveys during one or two years at a small group of islands within the Bonaparte Archipelago (Fig. 1). Several hundred colonies from around 60 species and 15 genera were sampled during autumn or spring season, with sampling focusing on species of *Acropora* (Tables 1 and S2). Of the total number of *Acropora* species know in the region, approximately 90% were sampled in spring and 40% in autumn, compared to 30 and 4% of non-*Acropora* species, respectively. Inferences about spawning during these seasons were drawn from in situ or microscopic examination of pigmented eggs within colonies, and there are no observations of coral spawning for the inshore Kimberley reefs.

The main season of spawning on inshore Kimberley reefs is probably during autumn, but with second multi-specific spawning also occurring during spring at a similar time to the oceanic reefs in the region (Tables 2 and S2). Of the species of *Acropora* sampled in spring ($n = 35$) and autumn ($n = 16$), 42% were inferred to spawn in spring and 87% in autumn (Table 2). Of the 60 common non-*Acropora* species, there was evidence of only 5% spawning in spring and 7% in autumn. The low proportion of non-*Acropora* spawning at these times suggests reproductive activity outside the peak spring and autumn spawning windows by these taxa, and/or is a consequence of low replication and a possible split-spawning. Although not observed in situ, spawning by a few species of Mussidae and Faviidae in aquaria at Kimberley Marine Research Station (KMRS) at Cygnet Bay occurred at a similar time as at the oceanic reefs during two years, 7–9 nights after full moon in March (A. McCarthy & A. Heyward, 2012, personal communication). There is currently no evidence of spawning in the inshore reefs of the Kimberley occurring a month earlier than on the oceanic reefs, as tends to occur on

parts of the Great Barrier Reef. If this was to occur in the Kimberley, spawning on the inshore reefs would be expected in February or March in autumn, and September or October in spring. Although sampling has not been conducted during these months, the existing data demonstrate that spawning did not occur exclusively a month earlier than on the oceanic reefs and that multi-specific spawning events have also occurred later in the season, during April in autumn and November in spring. Evidence for late spawning during autumn and spring may reflect a split-spawning during the years of sampling, as on the oceanic reefs.

Of 31 species sampled from seven genera on the inshore Kimberley reefs during late March, 30 had pigmented eggs and were likely to spawn in early April. This included many species that were sampled with low ($\leq 5$ colonies) replication, indicating that autumn is the main season of spawning. Indeed, based on the timing of the full moon and spawning on the oceanic reefs, the autumn spawning during that that year (2007) was likely spilt; so many colonies and species may have also spawned in early March, providing further evidence for autumn being the primary season of spawning for the region. Of 63 species sampled in late October, 25% contained pigmented eggs and were likely to spawn in early November, of which the majority were *Acropora*; 37% of the 35 species of *Acropora* contained pigmented eggs. However, eggs were absent from many of the colonies sampled with low replication ($\leq 5$ colonies) and the spring spawning may have been split, based on the timing of the full moon and the data for the oceanic reefs. Consequently, a proportion of colonies and species probably spawned in early October and future work may identify a higher proportion of species and colonies participating in a spring spawning. It remains to be determined whether the inshore reefs of the Kimberley display a similar degree of spawning synchrony during any one month in autumn and spring as on the oceanic reefs, or whether inshore spawning is more protracted over several months with seasonal peaks around autumn and spring, as may be the case on Indonesian reefs to the north (*Baird, Guest & Willis, 2009*). There are few data for the non-*Acropora* corals, which are most likely to have less synchronous patterns of spawning, and nor are there currently any data for brooding corals that are probably common throughout parts of the region. The brooding corals in the Kimberley are likely to display similar patterns of reproduction to those at the oceanic reefs, with planulation occurring during many months through spring to autumn, and perhaps extending into some winter months.

## Regional patterns of coral reproduction: Pilbara

There are extensive near-shore and mid-shore reefs systems throughout the Pilbara. Within the region much of the available information exists for the Dampier Archipelago (e.g. *Blakeway & Radford, 2004*; *Griffith, 2004*; *Marsh, 2000*; *Richards & Rosser, 2012*; *Veron & Marsh, 1988*) and there is less information for reefs in the west Pilbara (but see *Marsh, 2000*; *Richards & Rosser, 2012*; *Veron & Marsh, 1988*). The general pattern of coral diversity is similar throughout the Pilbara, with between 200 and 230 species recorded at the Dampier Archipelago, and at the mid-shore Montebello and Barrow

Island reefs. A slightly higher number recorded at the Dampier Archipelago may be due to greater diversity of habitats and environmental conditions (*Griffith, 2004*; *Marsh, 2000*; *Richards & Rosser, 2012*). However, there are distinct assemblages of coral species among the inshore reefs and those throughout the archipelago, reflecting the cross-shelf variation in environmental conditions and habitat types (*Blakeway & Radford, 2004*; *Richards & Rosser, 2012*). Average total hard coral cover for the inshore reefs of the Pilbara is approximately 20%, with the dominant families Faviidae and Dendrophylliidae having contributed to much of this cover (*Speed et al., 2013*). However, coral community composition can also vary dramatically among the inshore reefs and species of *Acropora,* Faviidae, *Platygyra, Turbinaria* and *Pavona* are common in some communities (*Blakeway & Radford, 2004*). The outer reefs of the west Pilbara can have communities characteristic of clearer water, with approximately twice the coral cover and a higher diversity. In particular, within the back-reef habitats many massive *Porites* colonies are associated with extensive coral assemblages, including a high cover (> 50%) of *Acropora* (*Marsh, 2000*; *Speed et al., 2013*).

Coral reproduction in the Pilbara region has been investigated at several reefs, with over 1,000 colonies sampled from 115 species, during one or more years (Table S1). Of the total number of *Acropora* species know in the region, 71 and 88% were sampled in spring and autumn, respectively, compared to 28 and 46% for the non-*Acropora* species (Tables 1 and S2). By far the majority of these data were from the Dampier Archipelago, and the times of reproduction were inferred from in situ ranking of gamete development, microscopic investigation of egg sizes and histological analyses of some spawning and brooding corals. Spawning has also been observed in situ on several occasions. Given the frequency and timing of disturbances to Pilbara reefs in recent years, including dredging operations, temperature anomalies and cyclones, some data from the region were probably biased by coral colonies having insufficient energy reserves to invest in reproduction. In these instances, the proportion of species and colonies reproducing could be underestimated.

The first discovery of coral spawning in Western Australia was in the Dampier Archipelago (*Simpson, 1985*). Early research showed corals spawning exclusively in autumn over two consecutive years, in 46 species of coral from seven families. The presence of mature eggs in some non-*Acropora* species after the main spawning event indicated split-spawning over two consecutive lunar cycles, but there was no evidence of spawning during spring. Subsequent research has documented multi-specific spawning by a small proportion of colonies and species during spring (October–December). Within the Dampier Archipelago, a small number of tagged colonies seemed to spawn consistently either in autumn or in spring and have only one gametogentic cycle. Of the species of *Acropora* sampled in spring ($n = 35$) and autumn ($n = 43$), 34% were inferred to spawn in spring and 98% in autumn (Tables 2 and S2). Of the 69 common non-*Acropora* species, 43% spawned in autumn and one spawned in spring, although few were sampled in spring. Among the non-*Acropora* species, only *Favites flexuosa,* and possibly *Favites pentagona* and *Montipora undata* are thought to spawn in spring or early summer, while the proportion of colonies within species of *Acropora* known to spawn

during spring is generally low (< 20%) (Tables 2 and S2). Sampling around a split-spawning and with environmental stress has potentially underestimated the participation by corals in the spring spawning (October–December), but the primary spawning period is certainly autumn (usually March).

Many Pilbara reefs are dominated by corals such as massive *Porites*, *Pavona decussata* and *Turbinaria mesenterina*, which display different patterns of reproduction to most hermaphroditic species that participate exclusively in the spring and/or autumn spawning events. Within the Dampier Archipelago, repeated histological examination showed that these three taxa were gonochoric. Spawning occurred predominantly in December in the massive *Porites* (mainly *P. lobata*), as on the Great Barrier Reef (*Harriott, 1983a*; *Harriott, 1983b*). For *Pavona decussata*, spawning occurred during March and April, possibly due to split-spawning during that year (2007). In *Turbinaria mesenterina*, spawning occurred over several months, possibly from November–April. While *T. mesenterina* retained eggs after this period, this does not indicate imminent spawning as this species has been reported to have a gametogenic cycle of more than 12 months (*Willis, 1987*). While spawning has not been observed, frequent sampling of *P. lutea* demonstrated that it spawned during spring tides predominantly 3 days (2–4 days) after the full moon, in contrast to the usual times of spawning during neap tides approximately one week after the full moon. In addition to these spawning corals, the main periods of reproductive output for the brooding corals in the Pilbara are also likely to occur at times other than during the dominant spawning periods in autumn and spring. Although cycles of gametogenesis in brooding corals have not yet been investigated in the Pilbara, they probably culminate in the release of planula larvae over several months through spring to autumn, and possibly into winter months.

## Regional patterns of coral reproduction: Ningaloo

Ningaloo is an extensive fringing reef system almost 300 km in length, with diverse coral communities and over 200 species of hard corals from 54 genera (*Veron & Marsh, 1988*). Mean coral cover can be as high as 70% at areas of the reef flat and reef slope, but is typically less at other habitats such as in the lagoon (*Speed et al., 2013*). The remaining benthic cover is composed of coralline and turf algae, seasonal macroalgae growth and other benthic organisms. Within the coral communities, the Acroporidae are often most abundant, but the Faviidae, Poritidae, Pocilloporidae and soft corals are also common (*Speed et al., 2013*; *Veron & Marsh, 1988*). The deeper lagoons typically contain massive *Porites* bommies and patches of staghorn *Acropora*, while the outer-slope is dominated by robust corals with massive and encrusting growth forms, often *Platygyra sinensis* and prostrate *Acropora* (*Wilson, 2013*).

There is detailed reproductive data for some species at one location at Ningaloo and a comparatively poor understanding of spatial variation across this extensive system (Table S1). Coral reproduction has been investigated during several years, for several hundred colonies from 42 species and 11 genera (Table 1). Of the total number of *Acropora* species know in the region, approximately 44% were sampled in spring and 67% in autumn, compared to 14 and 20% of non-*Acropora* species, respectively.

Most data exist for several species of Acroporidae and Faviidae sampled during one or more months from spring to autumn at Coral Bay. Early work at Ningaloo suggests some variation in the time of spawning may exist among locations, with a higher proportion of corals spawning in March in the north and in April to the south, but this may also have been a consequence of split-spawning. Nonetheless, the studies of coral spawning at Coral Bay provide detailed information about temporal variation in spawning among months, lunar cycles, and the nights of spawning in autumn. Inferences about spawning times were drawn from in situ ranks of gamete development and microscopic investigation of egg sizes in random population samples and by re-sampling individual colonies, in addition to direct observations of spawning in situ.

Mass spawning at Ningaloo occurs during autumn, with a more protracted period of spawning over consecutive months, and little or no multi-specific spawning during spring (Tables 2 and S2). Of the species of *Acropora* sampled in spring ($n = 17$) and autumn ($n = 26$), 12% were inferred to spawn in spring and 92% in autumn, with one spawning exclusively in summer. Of the 69 common non-*Acropora* species, none were reproductively active in spring, compared to 20% in autumn (Tables 2 and S2). However, a low proportion of species (< 20%) and particularly colonies have been sampled during spring. Additionally, there are very few reproductive data from parts of Ningaloo other than Coral Bay. Most Acroporidae and Faviidae colonies at Coral Bay participated in mass spawning during a single month in autumn, but a small proportion of many species also spawned during other months through summer and autumn. Species typically spawned during one or two consecutive months, with no evidence of spawning during discrete months or of a multi-specific spawning during spring, as on northern reefs. There are numerous observations of slicks of coral spawn during spring, but the extent to which these are a product of multi-specific spawning remains unknown (R. Babcock & D. Thompson, 2015, personal communication). Within species, individual colonies had a single gametogenic cycle and usually spawned within a few consecutive nights. The mass spawning usually occurred during neap tides in late March or early April, 7–10 nights after the preceding full moon, but a small proportion of colonies of several species also spawned following the full moon or the new moon during months either side of the mass spawning.

Within the species spawning during autumn, most of their colonies (60–100%) participated in the mass spawning in early April following the full moon in late March, but during other years mass spawning occurred in the last week of March following an earlier full moon in March. Around the quantified mass spawning events in early April, a relatively small (< 20%) proportion of colonies from most species also spawned a month earlier (early March) or later (early May), following the preceding full moon or new moon, particularly in the non-*Acropora* species. A higher proportion (10–20%) of these colonies spawned during March than in May (< 10%), which may be due to a split-spawning during the years of sampling or may be typical of a more protracted spawning at Ningaloo. Early observations suggest that split-spawning is a common feature at Ningaloo, but whether it occurs during the same years and involves a similar proportion of species and colonies as on reefs further north remains to be determined.

Cooler waters at Ningaloo could result in slower rates of gametogenesis and an increased likelihood of split-spawning during years in which a full moon falls early in March, and/or a higher proportion of colonies participating in an April spawning than on northern reefs.

There was little evidence of spawning at Ningaloo during months other than in autumn. Less than a few percent of colonies of *Goniastrea retiformis*, *A. humilis* and *A. papillarae* had visible eggs in October, but none were pigmented and the times of spawning were unknown. Existing data suggests that *A. papillarae* is the only species that does not participate in mass spawning and spawns exclusively during summer, probably during December and/or January. Additionally, a small proportion (< 5%) of *Echinopora lamellosa* also spawned during summer in February, but with a higher proportion spawning during March (≈ 13%) and particularly April (≈ 80%). There are currently no data for species of corals such as massive *Porites* known to spawn during summer at other reefs throughout WA. Given that spawning seems to be more protracted at Ningaloo, future work may identify a higher proportion of species and colonies spawning during summer, particularly for the non-*Acropora*. There is also no existing information for the times of planulation in brooding corals at Ningaloo, but planula release is likely to occur at similar times to other northern reefs, from spring through to autumn, with perhaps a lower incidence in spring due to the cooler water temperatures.

## Regional patterns of coral reproduction: Abrolhos Islands and Shark Bay

The Houtman Abrolhos Islands have the highest latitude coral reefs in Western Australia. The coral communities are scattered among four islands, situated < 100 km from the coastline but near the edge of the continental shelf, with over 180 species from 42 genera of corals (*Veron & Marsh, 1988*). Coral cover ranges between 35 and 85% among habitats (*Dinsdale & Smith, 2004*), with an average cover for the region of approximately 44% (*Speed et al., 2013*). Unlike studies on a comparable latitude on the east coast of Australia (*Harriott & Banks, 2002*), the Abrolhos maintains high percentages of tabulate and particularly staghorn *Acropora* (*Abdo et al., 2012*; *Dinsdale & Smith, 2004*). Much of the remaining substrata were covered in turf and coralline algae, although patches of macroalgae are also common. Situated to the north of the Abrolhos Islands, Shark Bay is a large shallow bay (~12,950 km$^2$) with an average depth of 9 m and is enclosed by a number of islands (*Veron & Marsh, 1988*). The bay consists of vast seagrass meadows (*Wells, Rose & Lang, 1985*) and coral growth is restricted to waters with oceanic salinity, such as in the western side of the bay (*Veron & Marsh, 1988*), where 82 species from 28 genera of hard corals have been recorded (*Veron & Marsh, 1988*). Corals from the families Acroporidae and Dendrophylliidae are found in similar abundance of approximately 10–15% cover, and other genera found in low (< 2%) cover include *Montipora*, *Platygyra*, *Pocillopora*, and *Porites* (*Bancroft, 2009*; *Cary, 1997*; *Moore, Bancroft & Holley, 2011*; *Speed et al., 2013*).

Coral reproduction has primarily been investigated during 1 year at the Abrolhos Islands, around the predicted time of mass spawning in autumn (Table S1). Of the total

number of species know in the region, approximately 49% of the *Acropora* and 34% of the non-*Acropora* were sampled in autumn, but with no sampling at other times of the year (Table 1). Several hundred colonies from 107 species and 10 families were sampled in March 1987, and a small random sample of colonies during late February 2004 (Table S2). Most samples were from species of *Acropora* and Faviidae around the Wallabi group of islands. The times of spawning were inferred from in situ ranking of gamete development, microscopic investigation of egg sizes and stages, and direct observation of spawning in situ and in aquaria. In addition to random sampling, tagged colonies were re-sampled before and after the main nights of spawning.

There is clearly a mass spawning by a high proportion of many *Acropora* species at the Abrolhos Islands during autumn, but no knowledge of whether corals also spawn during spring or summer (Tables 2 and S2). Of the 107 species sampled, 58 species participated in the main two nights of spawning in March, with a further 36 species likely to spawn on other nights during March; a similar proportion of *Acropora* (49%) and non-*Acropora* (31%) participated in the March spawning. Spawning occurred primarily over the 10 and 11 nights after the full moon, during spring tides of small amplitude (< 2 m), with reports of other spawning events also between 8 and 11 nights after the full moon. In addition to the species and colonies that spawned over a few consecutive nights, there was also evidence of a more protracted spawning by many colonies and species over a greater number of nights, and possibly also during April and/or other seasons. Within the species of mass-spawners, the mean number of colonies participating was 70%, and ranged between 10 and 100%. Most species spawned over a few nights, but within the assemblage spawning was probably protracted over almost three weeks, as early as a few nights before the full moon and up to two weeks later. Additionally, gametes were absent from a variable proportion of colonies in approximately half the species observed to spawn in March, and from all colonies in an additional 13 species, suggesting they either did not spawn during that year or were likely to spawn during a different season. Slicks of spawn have also been observed at the Abrolhos in February, although subsequent sampling suggested the bulk of the community was likely to spawn in March. The species known to spawn during months other than March on more northern reefs were either not sampled, or had a proportion of colonies without eggs and were sampled in low replication. There is currently no reproductive information for brooding corals, which are likely to release planulae over several months from spring to autumn, but with perhaps a reduced reproductive window due to cooler water temperatures.

## Regional patterns of coral reproduction: Southwest Region

Within the temperate southwest region of WA, corals are near their geographical limit. Reefs where corals are known to occur include Rottnest Island, Hall Bank, and some patches of reef within lagoons adjacent to the Perth mainland, such as at Marmion and Jurien. Rottnest Island has the most abundant coral communities, with 25 species from 16 genera. *Pocillopora damicornis* dominates certain areas (*Veron & Marsh, 1988*), which is a consequence of clonal reproduction (*Stoddart, 1984*). Clonal reproduction may also be important for other species at Rottnest Island with more tropical affinities,

such as *Acropora* sp. and *Porites lutea* (Crane, 1999). Among the remaining corals, the dominant taxa are species of Favidiiae with subtropical affinities, such as *Goniastrea australiensis*. Macroalgae (*Sargassum* and *Ecklonia*) are common around Rottnest Island and contribute up to 60% of benthic cover (Wells & Walker, 1993). Between Rottnest Island and the Perth metropolitan coastline is Hall Bank, a small reef with a low diversity (14 spp.) but a high cover (≈ 50%) of corals, of which most are *Favites* and *Goniastrea* (Thomson & Frisch, 2010). In contrast, the reefs adjacent to the coastline have a lower coral diversity and cover, such as Marmion lagoon with 10 species from eight genera (Veron & Marsh, 1988). Fleshy macroalgae are dominant on most of the temperate reefs, but corals are can often be found among the algae in low density (Thomson et al., 2012). The most abundant coral on these reefs is *Plesiastrea versipora*, one of the Indo Pacific's most widespread corals, however it rarely reaches large sizes and other species tend to have higher cover (e.g. *Goniastrea* spp. *Montipora capricornis*).

Throughout the southwest region, coral reproduction has been investigated only at Rottnest Island during one or more years throughout the 1980s and (Table S1). At Rottnest Island, a total of nine species and > 600 colonies were sampled over multiple seasons, for months to years (Table 1). The majority of the sampling has been conducted at two sites, which includes consecutive sampling and spawning observations of colonies prior to spawning around summer and autumn from January–May. Histological analyses were also used to investigate reproduction in three species (*Pocillopora damicornis, Alveopora fenestrata* and *Porites lutea*) from December–April. Mature gametes were found in colonies of the most abundant spawning corals over several months through summer and autumn (Tables 2 and S2). Histological analyses revealed *Pocillopora damicornis* at Rottnest Island to be both a brooding and spawning coral. Gametes and planula larvae were common in colonies through summer to winter (December to early April), being most common in March, and rare or absent in winter.

The available data for the southwest region are only from Rottnest Island where spawning by the dominant species appears to occur through summer and or autumn months (e.g. *Goniastrea aspera, G. australensis, Montipora mollis* and *Symphillia wilsoni*), a pattern similar to that seen on the subtropical reefs of Australia's east coast. Some colonies have been observed to spawn around the time of new moon rather than full moon, such as *Symphyllia wilsoni*, and *Alveopora fenestsrata*. Among the other dominant coral species in the region, there appears to be an extended reproductive season of two or more months at different times of year for different corals; for example, in summer for *Pocilloproa damicornis*, in early autumn for *Turbinaria mesenterina*. The apparent staggering of reproduction among species between February and May suggests that there is a relatively low level of synchrony within the temperate coral communities, but with perhaps a higher degree of synchrony among some conspecific colonies in late summer (Table S2). Because the species composition and level of coral cover varies so markedly among coral assemblages in the southwest, there is little or no knowledge of spatial variation in community reproduction throughout the region. For example, *Plesiastrea versipora* is numerically the most common coral in the region and across southern Australia, yet its reproductive biology in temperate waters is still poorly understood.

It is recorded as a mass spawner on tropical reefs (Magnetic Island, *Babcock et al., 1986*; Taiwan, *Dai, Soong & Fan, 1992*), but did not spawn with other subtropical corals such as *G. australiensis* in Moreton Bay, on the east coast of Australia (*Fellegara, Baird & Ward, 2013*). There is no knowledge of the distribution and patterns of reproduction in brooding corals through the southwest region, with the exception of *Pocillopora damicornis* at Rottnest Island.

## DISCUSSION

### Summary of coral reproduction across Western Australia

The observed differences in reproduction among Western Australian (WA) coral reefs are due to their varying community composition, modes of reproduction, and the cycles of gametogenesis for coral species. The most obvious differences in community composition are the higher abundance and diversity of Acroporidae and massive *Porites* on offshore reefs and tropical reefs north of the Abrolhos Islands. Among the inshore reefs and those south of the Abrolhos Islands, species of Faviidae, Pocilloporidae, *Turbinaria* and/or *Pavona* are more common and there is a notable decline in the abundance and diversity of coral species (*Lough & Barnes, 2000*; *Speed et al., 2013*; *Veron & Marsh, 1988*). Beyond the effect of community composition, the modes of reproduction displayed by the different coral species distinguished their cycles of gametogenesis and times of reproductive output.

As on most tropical reefs around the world, the dominant mode of coral reproduction on WA coral reefs is broadcast spawning. Within a year, most individual corals have a single cycle of gametogenesis that culminates in spawning during one or a few consecutive nights each year. However, the times of spawning and the degree of synchrony among and within species vary among the different regions, with a latitudinal gradient in the spawning activity among seasons. The primary period of spawning on all WA reefs (apart from the southwest region) is in autumn, often culminating in the mass spawning of a relatively high proportion of species and colonies during March and/or April.

Successive studies have added to the list of species known to mass spawn during autumn, but also to the list known to participate in a second multi-specific spawning during spring (October and/or November) on many WA reefs. The existing data suggest that biannual spawning by communities during autumn and spring is a phenomenon that occurs with increasing frequency from Ningaloo Reef north. Although more intensive sampling is necessary to clearly establish a latitudinal gradient, synchronous spawning by multiple species and colonies in the spring spawning is highest on the Kimberley Oceanic reefs, decreases considerably on Pilbara reefs, and may not occur on Ningaloo reefs–there is only anecdotal evidence of multi-specific spawning at Ningaloo Reef in spring. Of the 17 species of biannual spawners on the Kimberley Oceanic reefs that were sampled most rigorously in the other regions of WA, all spawned in autumn and five during spring in the Pilbara, and all spawned in autumn and none during spring at Ningaloo (Table 2). In addition to the reduction in spring spawning with increasing latitude, spawning may also become more protracted over consecutive nights or weeks

around the mass spawning in autumn from reefs in the Kimberley to the Abrolhos Islands (Table 2), although more data are again required to confirm this pattern.

Within these seasons, there is a comparatively poor understanding of spatial and temporal variation in spawning times (months, weeks, time of day). Mass spawning occurs most commonly in March and/or April, and the multi-specific spawning in October and/or November, often varying according to the timing of the full moon. As with coral communities on the Great Barrier Reef, spawning on WA reefs can be split over consecutive months in autumn and spring, depending on the timing of the full moon. The phenomenon typically occurs every few years, but can also occur in consecutive years. The nights of spawning were typically inferred from the presence of pigmented eggs in colonies days to weeks before the predicted dates, with very few direct observations of spawning and limited sampling conducted after the event. There is certainly a peak in spawning activity (mass spawning) over a few nights each year on most reefs, but with a variable participation by colonies and species in this primary spawning event. Most commonly, mass spawning occurs during neap tides between approximately 7–12 nights after the full moon, usually in March and/or April, on all reefs but for those in the temperate southwest region. However, intensive sampling of colonies over days and weeks at Ningaloo Reef has also documented spawning around the time of the new moon, as can occur in some species on the GBR (*Babcock et al., 1986*). Whether this pattern reflects a more protracted spawning that is unique to Ningaloo Reef, or is a feature of other WA reefs remains to be determined.

Despite some brooding corals being widely distributed and abundant on many of WA's coral reefs (e.g. species of Pocilloporidae and *Isopora*), there is currently little information about their cycles of gametogenesis and times of planulae release. Within a year, brooding corals on WA reefs probably have multiple cycles of gametogenesis culminating in the release of planluae larvae over several months, similar to those on the GBR (*Harriott, 1983b*; *Harrison & Wallace, 1990*; *Kojis, 1986*; *Tanner, 1996*; *Wallace et al., 2007*). On the Kimberley Oceanic reefs, planulae were present within *Isopora brueggemanni* and *Seriatopora hystrix* during several months through spring to autumn. Brooding corals on other WA reefs probably have similar cycles of planulation, but for perhaps a shorter reproductive window on higher latitude reefs. The relative proportion of planulae produced during different months of the year and the nights of their release relative to the phases of the moon are unknown for all brooding corals on all reefs.

## Methods for assessing coral reproduction

Consideration of coral reproduction is often required by environment mangers where development activities are proposed on or near coral reefs; the principle being that if coral spawning and larval settlement are concentrated during a discrete period then the potential impacts from development works can be minimised. Rigorous sampling and interpretation of reproductive status in coral communities is needed well in advance to provide time for planning; sampling is also needed to continue throughout to confirm predictions about time(s) of spawning. In all cases, the accurate prediction of the timing,
magnitude and duration of coral spawning is vital given the logistical complexity of development operations and the cost of delays.

Many early studies of coral reproduction employed rigorous, and often complimentary, methods because so few data existed. The resulting publications provided a detailed description of the methodology and the assumptions on which conclusions were based. Attempts to quantify cycles of reproduction today require a good knowledge of this background literature, and particularly the limitations of the different approaches. The most relevant literature and methods for a particular study will depend on the questions to be addressed, the regions in which the reefs are found and the species to be investigated. However, to maximise the knowledge gained and minimise the biases from sampling effort, many publications should first be read and understood; for example: *Alino & Coll, 1989*; *Ayre & Resing, 1986*; *Done & Potts, 1992*; *Fadlallah, 1983*; *Fan & Dai, 1995*; *Fong & Glynn, 1998*; *Glynn et al., 1994*; *Glynn et al., 1991*; *Harrison, 1993*; *Harrison & Wallace, 1990*; *Heyward & Collins, 1985*; *Sakai, 1997*; *Sebens, 1983*; *Shlesinger, Goulet & Loya, 1998*; *Stoddart, 1983*; *Stoddart & Black, 1985*; *Szmant-Froelich, Reutter & Riggs, 1985*; *Szmant & Gassman, 1990*; *Wallace, 1985*. In the context of environmental management, we provide some comments on the experimental design and methodology used in coral reproductive studies in Western Australia, which also provides some background to a thorough reading of the existing literature.

### Community composition

For environmental management, information about coral reproduction is often required at the level of the entire community. Thus, there is a need to assess the composition of coral communities across the susceptible reefs, habitats and sites, in order to quantify the relative dominance of the species. As the species list of corals at tropical reefs can be extensive, a convenient cut-off point must be chosen. Therefore, we suggest that the species be ranked in terms of their contribution to total coral cover, and those making a cumulative contribution to most ($\approx 80\%$) cover across all communities of interest be chosen for assessment of reproductive behaviour. However, consideration must also be given to whether certain species, although low in relative abundance, play a critical role in ecosystem maintenance (e.g. keystone species).

### Taxonomic resolution

Coral taxonomy and the identification of species for sampling are problematic in virtually any study of tropical coral communities; the issue cannot be understated. Identification to the finest taxonomic resolution possible is always desirable; however, the suggested approach of quantifying seasonal reproductive patterns for dominant taxa would work equally well for higher taxonomic groups. For example, a more practical approach depending on the diversity of species and the taxonomic skills of the researchers would be to group species according to a higher taxonomic level (e.g. Genus, Family) and to also consider growth form (e.g. massive, branching, encrusting, corymbose) and reproductive mode (spawner, brooder). The advantage with this approach is that uncertainty around the identity of species is obvious, rather than records of incorrectly

identified species becoming entrenched in the literature. Such approaches are valid where the objective of management is to protect reef integrity by ensuring resilience of the coral assemblage at a functional level.

### Inferring spawning and timing of sampling

Sampling of the dominant corals must take place throughout the potential reproductive seasons in order to determine the relative magnitude of reproductive output throughout the year. A key factor in the logical process of determining whether or not spawning has taken place is the construction of a series of data points through time that demonstrates the development of gametes and their subsequent disappearance after spawning. Oogeneic cycles in spawning corals take several months, so in species know to spawn biannually (March, October) or over a protracted period (September–April) eggs will be present in the population during most months. There is no evidence of corals spawning during winter months, so detailed sampling in this period is not necessary. A sampling program to determine the proportion of species and colonies spawning or releasing planulae throughout the year should, however, span at least nine months from the start of spring to the end of autumn.

Preliminary sampling should be conducted monthly, and take into consideration the influence of the lunar cycles. Ideally, sampling on Western Australian reefs should occur approximately one week before the predicted night of spawning, providing the greatest amount of information on the timing of spawning based on characteristics of gamete development; more than a week and eggs may not yet be pigmented, while less than a week the chances of missing an early spawning increases. The optimal time of sampling will depend on the assemblage. The presence of mature (pigmented) eggs or larvae (in brooding species), and fully developed sperm, followed by their subsequent disappearance, is the best basis for making strong inferences about the timing of spawning. It is important to note that in many corals, particularly the Acroporidae, eggs may not be pigmented more than two weeks prior to spawning and that unpigmented eggs may also be spawned, highlighting the need for large sample sizes and for sampling to be conducted following spawning events. In other taxa, particularly some Faviidae, eggs may be pigmented for two months or more before spawning.

A single annual sample is a weak basis for inference, particularly when spawning is split or staggered, for species that have protracted spawning seasons, or for brooding corals. It is vital that accurate records of the exact timing of sampling are reported as metadata, in order for clear conclusions to be drawn regarding the timing of spawning based on sequential sampling.

In addition to re-sampling the assemblage through time, tagged colonies would ideally be resampled to strengthen inferences about the time(s) of spawning. This eliminates doubt about whether the presence or absence of gametes is due to a spawning event, or due to variation in the timing of spawning among colonies within a population; it is particularly useful for species that spawn biannually. Consideration must obviously be given to the number of samples that can be taken from a single colony, so as not to cause significant stress and divert energy investment away from reproduction. We suggest that samples from

individual colonies are therefore taken strategically, according to the wider pattern identified in the population from which random samples are also taken. For example, sampling an individual colony to determine whether it participates in both a spring and autumn spawning, or in both months of a split spawning, rather than during many months of the year.

### Sample size

Sample sizes must be adequate for the purposes of the study and to account for the background variation in reproduction among species within the community, among conspecific colonies during a year, and among years. If all colonies are reproducing and spawn during the same month, then the level of replication required is small–but considerable sampling is first required to establish this trend and it is uncommon for many species on most reefs. Relying on a fixed sample size for all species can become problematic when colonies spawn during different months, different seasons, if stressed colonies are not reproducing, if they have separate sexes, or if spawning during a year is split. Simulations carried out to assess the power of sampling to detect reproductively mature colonies in coral communities (*Styan & Rosser, 2012*) can provide useful guidelines for designing sampling programs, after the underlying assumptions have been reviewed and the background variance established in the context of the assumptions of the simulation. The required replication can range from a few colonies per species when all spawn synchronously over a few nights each year, to many more colonies for assemblages with mixed patterns of reproduction during some years. For example, on a reef when 30% of the assemblage is spawning in spring, many colonies per species will need to be sampled following the full moon in October during a year of split-spawning (after the October spawning) so as not to underestimate the significance of the event, especially if a proportion of colonies are not reproducing due to environmental stress. Otherwise, insufficient sampling would not identify the period as important and it may not be investigated in subsequent years when spawning was not split and colonies not stressed.

It is important to note that the absence of eggs in a colony provides few insights into broader patterns of reproduction, further highlighting the need for sufficient replication. At least 10 or more colonies per species are therefore needed for adequate quantification of reproductive patterns on WA reefs that do not mass spawn during a single month each year–however, the replication required on each reef can only be determined after background variation in space and time are first established. We argue that for most WA reefs it is better to first sample the most abundant species rigorously to determine their pattern of reproduction, rather than sample most species within low replication. Additionally, within colonies not all polyps may be reproductive, so multiple samples from single colonies are advisable. For example, where both in situ and microscopic examination of eggs are used to infer times of spawning in certain coral species (e.g. staghorn *Acropora*), eggs may be observed in one method but not the other.

### Use of existing data and streamlining of sampling

Studies based on the sampling design principles above are rare, not only in WA but globally due to logistical demands. However, they are necessary for environmental

managers because there is insufficient knowledge of the underlying reproductive biology of seasonality and within-population synchrony in coral species at any given location. Where there is well documented information on seasonality and synchrony, sampling may be streamlined. For example, when the species' annual gametogenic cycle has been described and it has been shown that the population spawned with virtually 100% synchrony during one lunar period of the year, sampling could be conducted immediately before and after the predicted spawning window. However, for most species of coral in WA such information is lacking, making more extended sampling periods necessary.

Once the community has been defined and the experimental design confirmed, methods that can be used to determine the time of spawning or planulae release include: spawning observations, recruitment to artificial substrata, in situ examination of gametes, microscopic examination of gametes (immediate and preserved), and histological examination of gametes. The most appropriate method depends on the question to be addressed, the region in which the reefs are found and species being investigated, but a rigorous assessment usually combines multiple approaches.

### Direct observations

Direct observations to establish the date and time of spawning include those made of colonies in situ or in aquaria (e.g. *Babcock, Willis & Simpson, 1994*). In situ observations are the most reliable way to confirm spawning, but are rarely conducted because the logistic difficulties limit replication. The most useful way to apply in situ spawning observations is therefore to combine them with data from previous reef surveys and in situ observation of gamete development (see below). Aquarium observations present similar logistical issues, and inflict some level of stress on colonies that potentially alters their time of spawning. The approach has been used more successfully in brooding corals kept in aquaria for several months, with the dates of planula release around lunar phases determined each day with the use of planula collectors (e.g. *Richmond & Jokiel, 1984*; *Jokiel, Ito & Liu, 1985*).

Another observational method used to provide information on the timing of spawning in coral communities is visual surveys for coral spawn slicks, usually the morning after a spawning event. While this method is useful for establishing that some spawning has occurred, the approach cannot provide information on the scale of the spawning and the origin of the slicks is unknown; the absence of a slick obviously provides no evidence of spawning having not occurred. While all of these methods provide information about the time of spawning in a sub-set of species, they alone are not sufficient to establish the community-wide patterns of reproductive seasonality required for the purposes of managing environmental impacts.

### Coral recruitment

Coral recruitment surveys can be used to inform the general timing of peaks in reproduction (months/seasons) (*Wallace & Bull, 1982*) but they do not precisely describe temporal variation in peaks in reproduction. This is in part because pre-settlement larval periods vary among coral, particularly spawning and brooding corals, and artificial

substrata must be deployed and retrieved in a set period (≈ few weeks) before and after each spawning; the period before is required to pre-condition substrata with algal communities and the period after is required for larval metamorphosis and calcification to occur. Deploying and retrieving substrata each month is logistically difficult and provides indirect, and relatively imprecise, information about coral reproduction. A more precise identification of spawning times is usually required by environmental managers. The link between coral reproduction and recruitment can also be decoupled by unknown rates of larval mortality, current speeds and directions (*Caley et al., 1996*; *Edmunds, Leichter & Adjeroud, 2010*; *Hughes et al., 2000*) and recruitment provides retrospective rather than predictive insights into the times of reproductive output.

### In situ examination of eggs

In situ examination of eggs is the most common and perhaps useful means of determining times of spawning, provided certain criteria are met. If knowledge of the proportion of colonies participating in a spawning event is required it is vital to know whether the species are hermaphroditic or gonochoric (separate sexes), as eggs will obviously be absent from male colonies. Eggs of gonochoric species are often small and relatively colourless (*Harrison & Wallace, 1990*). Even where gonochoric species produce large coloured eggs, testes will remain colourless or white and difficult to distinguish from the white skeleton of a coral fragment. The sex ratio of gonochoric species must be known if the presence of mature (pigmented) eggs in colonies is to be used to infer the proportion spawning. Thus, in situ visual examination of gonochoric species is more difficult and likely to lead to incorrect conclusions. It is also vital not to sample the sterile tips or the edges of a colony, which have grown subsequent to the initiation of gametogenesis in the rest of the colony (*Oliver, 1984*; *Wallace, 1985*). Furthermore, is it very important to examine multiple polyps within each sample, and multiple samples from each colony, as some polyps may be sterile or have low fecundity. These points apply to whatever method of oocyte examination is to be employed.

In situ examination of eggs is most useful for branching corals that have large (≈> 0.5 mm) pigmented oocytes prior to (< 2 weeks) spawning that can be easily identified in the field (e.g. *Acropora* spp.). Colonies are generally examined in situ several days prior to the predicted dates of spawning. Maturity is examined in the field by breaking off coral sections to expose oocytes (*Harrison et al., 1984*), and several sections should be examined if eggs are not initially observed. Oocyte pigmentation is often used as an indication of maturity and timing of imminent spawning. Egg colour varies with developmental stage from small unpigmented eggs (indicating spawning is still some months away), to large pigmented eggs. Importantly, the size of mature eggs and the degree and colour of pigmentation varies among species; when mature *Acropora* eggs are typically pink or red, whereas the Faviidae may be blue or green, the *Montipora* may be brown due to the presence of zooxanthellae, while other species may have cream eggs when mature (*Babcock et al., 1986*; *Harrison & Wallace, 1990*; *Heyward & Collins, 1985*; *Shlesinger, Goulet & Loya, 1998*). In *Acropora*, pigmentation may not occur

until two weeks or less before spawning, in some Faviidae eggs may be pigmented as much as two months before spawning, and species such as *Acropora* and massive *Porites* have been observed to spawn unpigmented eggs (*Babcock et al., 1986*; *Harrison et al., 1984*; *Harrison & Wallace, 1990*; *Mangubhai & Harrison, 2007*; *Stoddart, Stoddart & Blakeway, 2012*), all complicating the use of egg colour as an indicator of imminent spawning. While in situ visual observations have led to many useful inferences about the timing of spawning in coral communities, they can be ambiguous and are best used as part of a sampling program that also uses microscopic examination of eggs, and ideally the sequential sampling of tagged colonies. Spawning times should also be confirmed by the disappearance of eggs from colonies following the predicted date of spawning.

### Microscopic examination of gametes

In situ examination of polyps is well supported by microscopic examination of the samples on the same day. Such examinations should be conducted on broken sections of coral under a dissecting microscope or with a hand lens, and can reveal gonads that were not visible underwater. This is particularly useful for species with small polyps and gonads, or those with a low fecundity (e.g. branching *Acropora*). Inferences about the times of spawning are also improved by investigating the developmental stages of testes. When testes are prepared and examined with a compound microscope (40× objective), sperm shape and motility can be observed. Testes enlarge markedly and sperm develop tails during the last month before spawning. Sperm heads remain spherical until the last one to two weeks before spawning, when they will become cone or acorn shaped, and a high degree of sperm motility occurs a few days before spawning (*Harrison et al., 1984*). Microscopic examinations are more time consuming, but always more reliable and informative than field observations alone.

Microscopic examination of gamete development can also be conducted on well preserved samples, but only after the sample has been decalcified with acid. Egg colour and shape are not retained following preservation and it is not possible to discern aspects of sperm morphology or behaviour. Where the dimensions of mature eggs or testes are known for species being sampled, measurements of their size can be can be used to make inferences about the likely time of spawning. Several studies have quantified the size of eggs within replicate colonies of a species at the time of spawning, and there can be variation in egg sizes within colonies and among conspecific corals (*Gilmour et al., 2016*; *Heyward & Collins, 1985*; *Mangubhai, 2009*; *Mangubhai & Harrison, 2007*; *Stobart, Babcock & Willis, 1992*; *Wallace, 1985*; *Wallace, 1999*). Egg size prior to spawning can also vary considerably from year to year, and is not a reliable metric alone for determining the month of spawning. Therefore, egg size can be used to estimate level of maturity and as an indicator of spawning with accuracy of perhaps two months, but probably not for a single month or less. Consequently, investigation of the preserved gametes is particularly useful for tracking their development over several months leading up to a spawning event, but to determine the month(s) spawning also requires in situ and microscopic examination as part of the sequential sampling program.

### Histological examination of gametes

Histology is used for corals that are not well suited to field examinations, usually due to their morphology and their having small polyps and eggs that are not easily visible with the naked eye (i.e. massive *Porites, Pavona*). Often these are also gonochoric species with separate male and female colonies, for which histology provides the only approach to describing the development of testes prior to spawning. This method is also commonly employed to assess reproductive status of brooding corals and the presence of planula larvae. Preparation for histological examination is time consuming and costly, and usually involves decalcifying tissue, dehydration and fixing of samples in wax, and then sectioning and mounting tissue to slides. Egg and testis development or presence of planula larvae can then be assessed using previous work as guides (*Szmant-Froelich, Reutter & Riggs, 1985*; *Vargas-Ãngel et al., 2006*). The development and growth of gonads and gametes can be tracked by measuring changes in size as well as morphological developmental features like sperm shape through time. As with in situ examination of gonads, sperm development stage is a particularly useful indicator of maturity and imminent spawning. Gamete development stages are frequently used in describing the reproductive status of corals sampled using histological methods, and also occasionally for microscopic examinations of freshly sampled tissues. Gamete development staging should be done with reference to published and accepted staging criteria available in the peer reviewed literature, and clearly defined so that the unambiguous interpretation of staging by others will be possible.

### Complementary methods

The most informative studies of coral reproduction involve sequential sampling of colonies using a combination of complementary methods. For example, determining the times of spawning may involve monthly examination of eggs in preserved samples. As eggs approach a size in which spawning is likely, then preserved samples may be supplemented or replaced by in situ and microscopic examination of egg size and pigmentation, as well as the size of testes and the stages of sperm development. When the night of spawning is predicted within a given month, the in situ and microscopic examinations of eggs and sperm are continued on a daily basis around the predicted nights of spawning, possibly supplemented by in situ observations of spawning and evidence of spawn slicks the following day. On all reefs, a proportion of species and colonies typically spawn over several nights, so continued sampling is required to quantify the proportion of colonies without eggs and to identify the main night of mass spawning within a period.

Great care should be taken to correctly record metadata and to correctly identify the main night of spawning relative to the full moon, particularly to note instances when split-spawning has occurred. Split-spawning typically occurs every few years and can occur over consecutive years, and not recognising the phenomenon has biased the existing data for Western Australian reefs. Also, supplementing these methods with tagged colonies that are sampled periodically through time will give a better indication of whether split-spawning has occurred, whether assemblages spawn biannually or

over a protracted period, or whether species have overlapping gametogenic cycles (e.g. *Turbinaria*). For gonochoric species with small polyps and gametes, or brooding corals, this sampling approach will likely involve the use of histological analyses as well as in situ and microscopic methods. These approaches are examples of applying multiple complimentary methods, but the best approach will depend on the aims of the study, the region, habitat, and species being investigated.

## Quantifying temporal variation in coral reproductive output

Since the discovery of mass coral spawning on the Great Barrier Reef, the phenomenon of mass- or multi-specific spawning has been documented at an increasing number of reefs around the world (*Baird, Guest & Willis, 2009*; *Harrison, 2011*). In many of these studies sampling has been conducted around the main periods of spawning on the nearest reefs and focused on species of *Acropora*, which are most likely to spawn synchronously and are easiest to sample. The results have established the timing and participation by species in the primary spawning event(s). However, fewer studies have provided detailed information about cycles of gametogenesis in spawning corals and planulation in brooding corals during other months of the year, despite a proportion of colonies of some of the most common species not participating in the primary spawning event(s) on a reef. For example, on inshore reefs of Western Australia's Pilbara region the *Acropora* may be relatively rare (≈ 5% by cover), whereas species of massive *Porites, Turbinaria* and *Pavona* that have novel cycles of reproduction may be among the most common (10–20% by cover) (*Baird et al., 2011*; *Stoddart, Stoddart & Blakeway, 2012*). Even on reefs where the *Acropora* are among the most common genera (25% by cover), such as on the oceanic reefs of the Kimberley, other common groups of corals such as the massive *Porites* (20%), *Isopora* (14%) and Pocilloporidae (10%) also have different reproductive modes or cycles.

An accurate assessment of the significance of periods of reproductive output requires knowledge of the proportion of colonies within each species releasing gametes or larvae during many months of the year. Additionally, several years of data with varying environmental conditions are required to understand the drivers of inter-annual variation, such as whether a low participation in a spawning event was due to environmental stress or split-spawning. Without these comprehensive data, surveys during previous years provide few insights into future spawning events, requiring substantial sampling effort to be repeated prior to every period of interest. In the worst instances, focusing only on the participation by species in a single month risks perpetuating a paradigm of mass spawning or missing a significant period of reproductive output.

A lack of accurate and unbiased information about times of reproductive output by coral communities impedes management initiatives aimed at reducing pressures to their early life history stages. Managers would ideally be provided with quantitative estimates of the reproductive output during different weeks, and even nights of the year, and its contribution to the long-term maintenance of populations. Obtaining this knowledge is logistically impossible until methods are developed that can easily quantify larval production, survival, connectivity, and per capita rates of recruitment per adult,

for the most abundant species within a community. However, with far less effort it is possible to obtain relative estimates of reproductive output for coral communities in different months of the year. Mass spawning was originally defined as '...the synchronous release of gametes by many species of corals, in one evening between dusk and midnight...' (*Willis et al., 1985*), taking place within a mass spawning period of up to a week following full moon on the Great Barrier Reef. There has since been debate about what constitutes a 'mass spawning' or a 'multi-specific spawning' on a reef, and a quantitative estimate of spawning synchrony has been developed (*Baird, Guest & Willis, 2009*) to assess biogeographic variation in spawning synchrony among species. Here we consider one approach to quantifying the significance of periods of reproductive output for coral communities on a reef, which combines the relative abundance of coral groups with the proportion spawning or releasing larvae during different months.

To apply information on reproductive synchrony to the management of ecological processes, such as reproduction and recruitment in coral communities, it is necessary to quantitatively weight this information according to community composition. Community composition on a reef varies considerably among habitats (lagoon, reef flat, reef crest, reef slope) and among sites within these habitats, particularly on inshore reefs. The abundance of corals with different cycles and modes of reproduction also vary among these habitats and sites, so careful consideration must be made of the assemblage of corals that best characterises the reef when assessing its times of reproductive output (or the assemblage for which reproductive information is required). Here we define the community based on percentage cover data, which is regularly collected during monitoring programs across habitats and replicate sites. These data provide a less biased sample of the community composition than is often obtained during reproductive surveys and can provide unexpected insights into which coral groups are most common and whether their patterns of reproduction are well known. For example, knowledge of coral reproduction is often for the most conspicuous and easily sampled species of *Acropora*, whereas little reproductive data may exist for other species of spawning (e.g. *Montipora, Porites*) and brooding corals (e.g. *Isopora, Seriatopora*) that may have a similar relative abundance on a reef.

Once the community composition has been quantified, a decision must be made about whether all taxa are to be included, or whether sampling effort can be reduced by excluding the rare corals. Although there are typically over tens of genera and hundreds of coral species on tropical coral reefs, many species are rare. For example, on Kimberley oceanic reefs there are over 35 genera and 300 species of hard corals, but 10 genera contribute approximately 85% of the coral cover and five genera contribute approximately 65% of the cover. On reefs with less diversity, comparatively few species may contribute much of the coral cover. A 'community' may therefore be defined by the corals that make up most (e.g. > 80%) of the total coral cover. An alternative to using a threshold of relative abundance is applying multivariate analyses to quantify the coral groups that best characterise, or distinguish, community structure through space and time, depending on the objectives of the study. Focusing on the detailed patterns of reproduction in the dominant species on a reef significantly reduces the sampling effort

required to quantify temporal variation in reproductive output, as finding and sampling rare species with sufficient replication is most time consuming. Care must obviously be taken in deciding the threshold for including dominant species in a 'community' and the means by which they are categorised (e.g. family, genus, species, growth form), which will need to be reviewed as communities change and as more reproductive data are obtained.

Further confounding assessments of the significance of reproductive periods on a reef is the identification of coral species. The issue is improved by considering only the most common species, but even these can be very difficult to distinguish. Variation in physical conditions among reefs and habitats, hybridization, reproductive isolation, and cryptic speciation make it impossible to correctly identify all of the colonies sampled during extensive reproductive surveys, and very few people are capable of correctly identifying most colonies correctly in situ. For example, many species of *Porites*, Faviidae, *Montipora* and *Acropora* are common on most tropical coral reefs, but within each of these taxa are many species easily confused in situ, even by experts. Errors in the identification of species will affect estimates of the number of species and the proportion of their colonies participating in a spawning, yet this can be the criteria by which the significance of spawning events is assessed. Inconsistent identification of species likely accounts for some apparent discrepancies in the times of spawning by species within regions of Western Australia presented here.

It is also impossible to correctly identify all species when quantifying the composition of coral communities in monitoring programs, particularly from photographic or video stills. Broader taxonomic, morphological and life history groups are usually used in this context. Thus, when quantifying the significance of spawning events for coral communities, errors can be avoided and efficiency increased by grouping some species to a higher taxonomic level (Genera/Family), but also distinguishing these according to growth form (e.g. branching, corymbose, encrusting; *Wallace, 1999*; *Veron, 2000*) and reproductive mode (spawner, brooder; *Harrison & Wallace, 1990*; *Baird, Guest & Willis, 2009*). The approach is obviously not needed for species that are abundant and/or functionally important on a reef and easily identified (e.g. *Isopora brueggemanni*, *Seriatopora hystrix, Diploastrea heliopora*).

Once the community of interest is defined and the relative cover of coral groups determined, the patterns of reproduction must be accurately quantified (see Results). The approach suggested here is to combine the community abundance and reproductive data to quantify relative estimates of reproductive output by the community throughout the year; it is therefore necessary to have both types of data for the same groups of corals. For example, in a hypothetical coral assemblage of two species, species A makes up 20% (relative) of the total coral cover and spawns only during October, and species B makes up 80% of the coral cover and spawns during March. Reproductive output for the community is therefore 20% during October and 80% in March. In this example, both species reproduce exclusively during a single month, whereas communities characteristically have species that spawn during two or more months a year, due to phenomena such as split-spawning, asynchronous spawning and within-population

biannual spawning. Additionally, brooding corals release larvae over several months within a year. Here we use available data to apply the approach at two *hypothetical* reefs with contrasting coral assemblages and patterns of reproduction (Fig. 2), which are similar to those at some oceanic and inshore reefs of Western Australia. The method for calculating reproductive output during each month is simple (Table S4), but considerable sampling effort is required to produce accurate estimates of community composition and the proportion of colonies within each taxa reproducing each month (see Results; Table S4). Quantifying the proportion reproducing each month will usually require monthly sampling, or at least bimonthly, although less effort may be required after several years under a range of conditions and depending on the methods used.

In a hypothetical example for an oceanic and inshore reef at north-western Australia, the monthly reproductive output differed according to their community composition and the cycles of reproduction within and among coral taxa (Fig. 3). The main month of reproductive output at both reefs was March, but with more synchronous spawning in March at the oceanic reef and more protracted spawning over March and April at the inshore reef. At both reefs, over 40% of the communities reproductive output occurred during other months, but for different reasons. The oceanic reef had a higher (≈ 20%) reproductive output during spring (October/November) and particularly October than at the inshore reef (≈ 8%), due to a higher number of species spawning biannually, a higher abundance of *Acropora*, and a tendency for spawning to be more synchronous during a single month. There was a much higher reproductive output in December (22%) on the inshore reef, due to the abundance many massive *Porites* that spawn predominantly in December. During several other months of the year reproductive output was higher on the oceanic reef, due mainly to the many massive *Porites* spawning and brooding *Isopora* releasing larvae from spring to autumn. In contrast, the brooding corals in this example were rare on the inshore reef and the spawning over several months was restricted to the *Turbinaria*. These estimates of reproductive output for the oceanic reef and the inshore reef are hypothetical, intended only to provide a worked example.

This is one of several possible approaches to quantifying temporal variation in reproductive output for an entire community throughout the year, intended to aid management decisions. The approach is aimed at identifying the months in which significant reproductive output occurs at the scale of the entire community, and more detailed temporal sampling within these months is required to determine the nights of spawning and planula release relative to the phases of the moon. In the context of managing environmental impacts, the approach presented here has several limitations. Most significantly, by considering the reproductive output of the community as a whole, it does not sufficiently recognise the significance of periods of reproductive output by functionally important species with unique cycles of reproduction. For example, massive *Porites* on inshore reefs that spawn predominantly just a few nights after the full moon in December (*Stoddart, Stoddart & Blakeway, 2012*), or species on the oceanic reefs that may spawn exclusively during spring (Table S4). Additionally, brooding corals may have a negligible (e.g. < 20%; Table S4) proportion of planula release during the month and

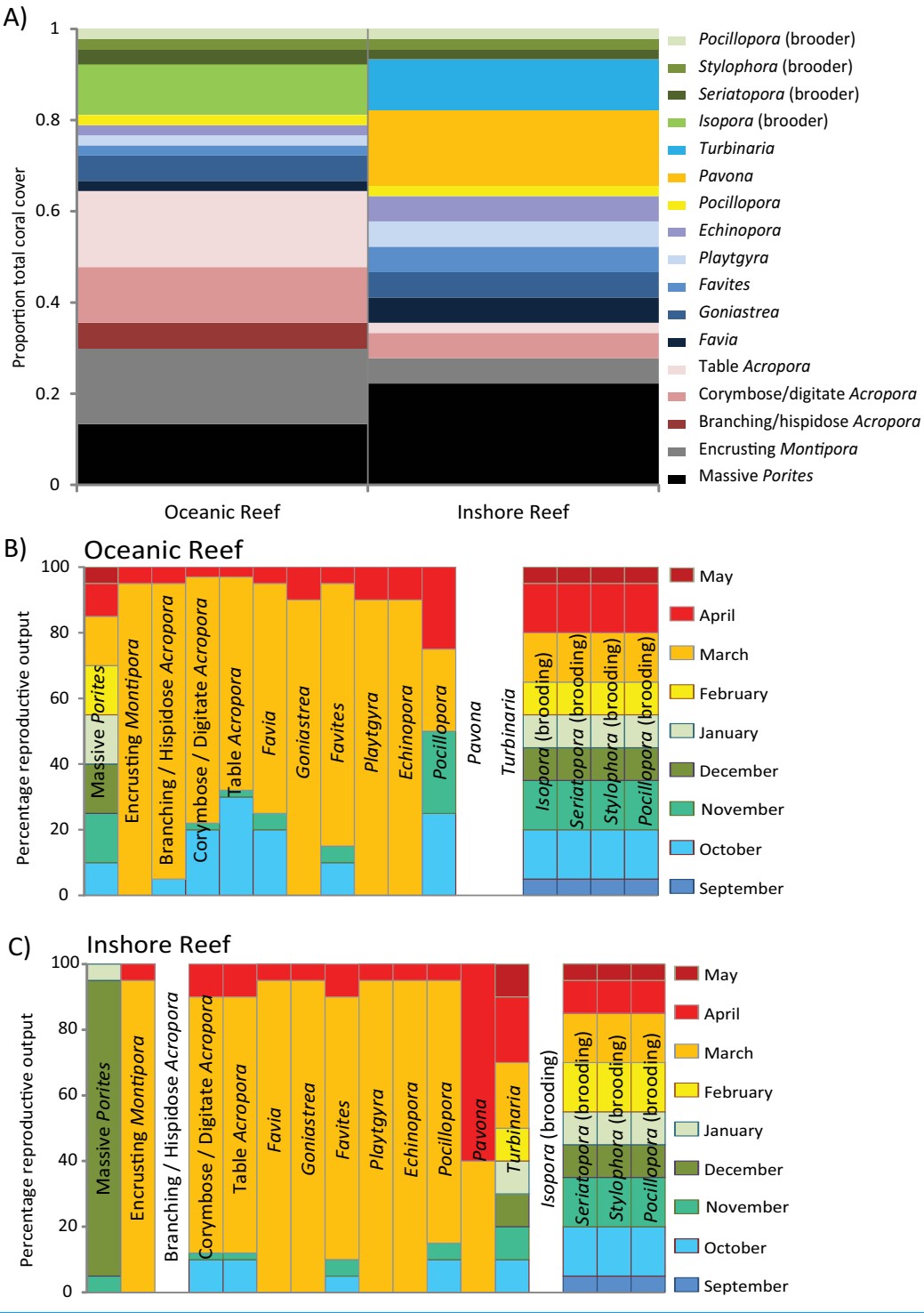

**Figure 2 Variation in composition and times of reproduction at Western Australian reefs.**
(A) Proportional contribution of coral groups to total coral cover at a hypothetical oceanic and inshore reef, and the percentage reproductive output (spawning, planula release) through the year at the (B) oceanic and (C) inshore reef. In this example, *Povona* and *Turbinaria* were absent from the oceanic reef and *Isopora* absent from the inshore reef.
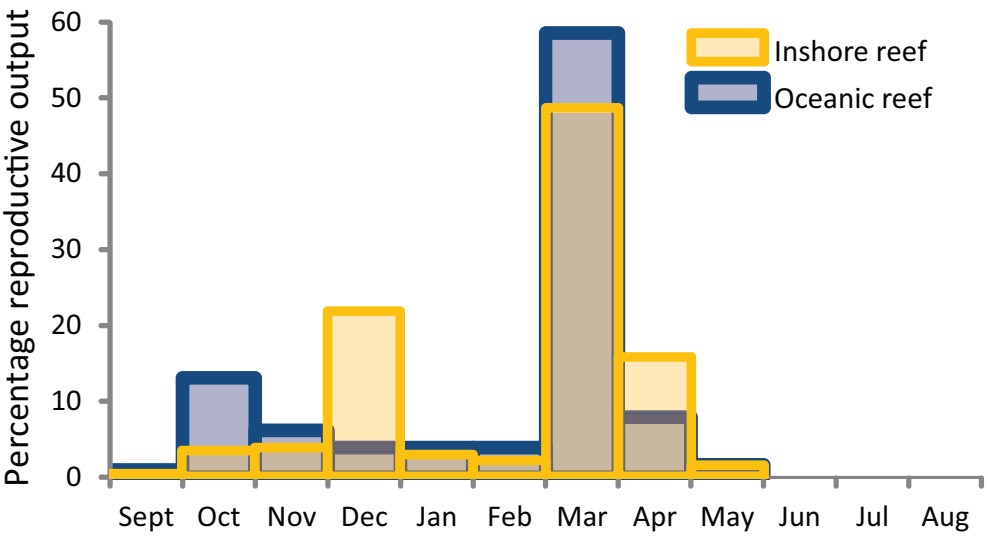

**Figure 3 Percentage reproductive output during each month on a hypothetical oceanic and inshore reef at north-western Australia.** Calculations are based on the relative abundance of coral groups within the community and the proportion reproductive output for spawning and brooding corals during each month of the year (Fig. 3; Table S4).

particularly main night of mass spawning on a reef, occurring around the full moon over several other months through the year.

Importantly, the proportion of reproductive output each month will vary among years according to changes in community structure and particularly the occurrence of split-spawning or environmental stress. Consequently, several years of data collection are required to obtain a reasonable understanding of reproduction on the reef before sampling effort can be reduced. For example, it may be concluded incorrectly that reproductive output was not significant during March, the usual month of mass-spawning by coral on most Western Australian reefs, if spawning was split (March/April) or because environmental stress (e.g. poor water quality, cyclone damage or mass-bleaching) precluded reproduction. Another limitation of this approach is that the temporal resolution is presented to the calendar month, with the assumption that spawning occurs approximately one week after the full moon, whereas reproductive output in some spawning corals (e.g. Faviidae at Ningaloo Reef) and many brooding corals are likely to occur around the new moon. Furthermore, affording protection to only the main month of mass spawning and not in other months may have unforeseen consequences, such as affecting connectivity between reefs following spawning events in which oceanographic currents differ (*Gilmour, Smith & Brinkman, 2009*), or by reducing the genetic diversity of new recruits. This highlights the need to consider reproductive output for the entire community in the context of more detailed reproductive data for abundant or functionally important species of corals. Assessing the strengths and weaknesses of the approach requires a dedicated sampling design, on reefs with different coral communities through several years of environmental conditions. Other (and possibly better) approaches exist; however we present one here to formally introduce and hopefully build the concept.

### Funding

This data synthesis was funded by the Western Australian Marine Science Institute (WAMSI), the Australian Institute of Marine Science (AIMS), and the Commonwealth Scientific and Industrial Research Organisation (CSIRO). The funders had no role in study design, data collection and analysis, decision to publish, or preparation of the manuscript.

### Grant Disclosures

The following grant information was disclosed by the authors:
Western Australian Marine Science Institute (WAMSI).
Australian Institute of Marine Science (AIMS).
Commonwealth Scientific and Industrial Research Organisation (CSIRO).

### Competing Interests

The authors declare that they have no competing interests.

### Author Contributions

- James Gilmour conceived and designed the experiments, analyzed the data, wrote the paper, prepared figures and/or tables, reviewed drafts of the paper.
- Conrad W. Speed analyzed the data, wrote the paper, prepared figures and/or tables, reviewed drafts of the paper.
- Russ Babcock conceived and designed the experiments, wrote the paper, reviewed drafts of the paper.

### Data Deposition

    The research in this article did not generate any raw data.

### Supplemental Information

Supplemental information for this article can be found online at http://dx.doi.org/10.7717/peerj.2010#supplemental-information.

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
