# Peer review of "Coral reproduction in Western Australia"

_PeerJ, doi:10.7717/peerj.2010_

## Round 0.1 · original submission · Minor Revisions

I have now received comments from two expert reviewers. Both reviewers were generally very positive about your submission, and only have quite minor suggestions and corrections. Having evaluated your manuscript myself, I agree with their comments, and hence, my decision is 'minor revision'. I look forward to receiving a revised manuscript.

Reviewer 1 ·

Basic reporting

This paper is clear and fairly well written. It summarizes the pertinent literature on coral reproduction in Western Australia and includes the "gray literature", making this information available to a wider audience. Because this is a review, the manuscript is fairly long. The figures are clear and relevant to the content. Two relatively minor points: The authors repeatedly use specie for a single species and in line 856, coral is spelled as coal.

Experimental design

The research here addresses a huge issue in conservation biology - how do we adequately assess reproduction in terms of environmental mitigation. This is an extremely important topic that has been alluded to in previous work, but this paper outlines a process to capture the essential elements critical for understanding impacts of human projects on coral reproduction. The paper also addresses significant aspects of reproduction that need to be considered in all studies of coral reproduction, whether for purely scientific study or for management. The authors have developed, and demonstrate how to apply, a method to quantify periods of time that are significant in terms of the contribution of major community members to reproductive output, and also highlight the concerns for "rare" species.

Validity of the findings

An extensive data set for reproductive parameters for Western Australian species has been collated and used for the development of a prototype model that environmental managers could apply to assess impacts on coral community reproduction. This model clearly could be applied in other regions of the coral world.

Reviewer 2 ·

Basic reporting

No Comments

Experimental design

No Comments

Validity of the findings

No Comments

Additional comments

This paper titled as “Coral reproduction in Western Australia” is a good review of coral spawning information in a large region of Western Australia, from tropical to subtropical reefs. Such information was collected from a large amount of papers including confidential reports to industry and government, and was well organized in figures and tables. I believe that this review makes easy to understand the regional characteristics of coral reproduction, and recommend that this paper is worth publishing in Peer J, except that some references are required for several sentences. Please see below for minor revisions.

L173-175: Is this information supported by any references?

L690: mangers to managers

L886-889: Do you have some data supported this information? Authors need to clarify a reference or data.

L913-915: Need references.

L970-971: Need references.

---

## Round 0.2 · accepted · Accept

The authors have answered the reviewers' comments well, and this new version is ready to be published.

I have noticed a few small things that need to be edited here and there, please ensure these are done at the proof stage or earlier if possible.
1. In Table 1, ensure region names are NOT in italics to distinguish from generic names.
2. Check spelling of "Damiper".
3. Check capitalization is consistent with "reefs", e.g. "Pilbara reefs" or "Pilbara Reefs".
4. Replace "some examples include (...." with "(e.g. LIST)".